**∂ | Open Peer Review** | Eukaryotic Cells | Methods and Protocols

# Strategies for genetic manipulation of the halotolerant black yeast *Hortaea werneckii*: ectopic DNA integration and marker-free CRISPR/Cas9 transformation

Yainitza Hernandez-Rodriguez,[1] A. Makenzie Bullard,[2,3] Rebecca J. Busch,[4] Aidan Marshall,[4] José M. Vargas-Muñiz[4,5,6,7]

**ABSTRACT** *Hortaea werneckii* is a halotolerant black yeast commonly found in hypersaline environments. This yeast is also the causative agent of tinea nigra, a superficial mycosis of the palm of the hand and soles of the feet of humans. In addition to their remarkable halotolerance, this black yeast exhibits an unconventional cell division cycle, alternating between fission and budding cell division. Cell density and the salt concentration in their environment regulate which cell division cycle *H. werneckii* uses. Although *H. werneckii* have been extensively studied due to their unique physiology and cell biology, deciphering the underlying mechanisms behind these remarkable phenotypes has been limited due to the lack of genetic tools available. Here, we report a new ectopic integration protocol for *H. werneckii* using polyethylene glycol-CaCl$_2$ mediated protoplast transformation. This approach relies on a drug (hygromycin B) resistance gene to select for successful integration of the genetic construct. The same construct was used to express cytosolic green fluorescent protein. Finally, we developed a marker-free CRISPR/Cas9 protocol for targeted gene deletion using the melanin synthesis pathway as a visual reporter of successful transformation. These transformation strategies will allow testing hypotheses related to *H. werneckii* cell biology and physiology.

**IMPORTANCE** *Hortaea werneckii* is a remarkable yeast capable of growing in high salt concentration, and its cell division cycle alternates between fission-like and budding. For these unique attributes, *H. werneckii* has gathered interest in research programs studying extremophile fungi and cell division. Most of our understanding of *H. werneckii* biology comes from genomic analyses, the usage of drugs to target a particular pathway, or the heterologous expression of its genes in *S. cerevisiae*. Nonetheless, *H. werneckii* has remained genetically intractable. Here, we report on two strategies to transform *H. werneckii*: ectopic integration of a plasmid and gene deletion using CRISPR/Cas9. These approaches will be fundamental to expanding the experimental techniques available to study *H. werneckii*, including live-cell imaging of cellular processes and reverse genetic approaches.

**KEYWORDS** *Hortaea werneckii*, transformation, morphology, ectopic integration, CRISPR/Cas9

H ortaea werneckii (*Dothideales, Ascomycota*) is a black yeast of particular interest due to its ability to grow in high salinity (1). This black yeast is commonly isolated from hypersaline environments, including seawater and solar salterns. Not only does this yeast display a remarkable halotolerance, but it also exhibits an unconventional cell division cycle (2). *H. werneckii* first grows in a pattern similar to that of fission yeast. After the first septation, it switches to budding from the poles. This division

Address correspondence to José M. Vargas-Muñiz, j.vargas.muniz@vt.edu.

The authors declare no conflict of interest.

pattern and cell morphology depend on the growth environment and cell density (3, 4). *H. werneckii* is also of clinical interest due to its ability to cause superficial mycosis of the hands and feet, known as tinea nigra (5–7). Tinea nigra frequently occurs in countries located in the tropics (6). On rare occasions, *H. werneckii* can cause a systemic infection known as disseminated phaeohyphomycosis (7). *H. werneckii* isolates exhibit great phenotypic diversity, including drug susceptibility and pathogenicity variation (3, 8, 9). This phenotypic diversity also correlates with the genetic diversity of *H. werneckii*, as *H. werneckii* isolates are either haploids or intraspecific hybrids of two divergent isolates (10–12). For these reasons, *H. werneckii* is an emerging model for understanding eukaryotic adaptation to hypersaline environments and how hybridization events contribute to fungal adaptation to extreme environments (9).

Most of our understanding of *H. werneckii* biology and adaptation to hypersaline conditions have been derived from genomics, metabolic, and physiological analyses (1, 9–22). However, the lack of reliable genetic tools has limited the ability to test mechanistic hypotheses related to *H. werneckii* physiology and cell biology. Different methods exist to genetically transform fungi, including protoplast-mediated, *Agrobacterium*-mediated, electroporation, generation of chemically competent cells using lithium acetate, and biolistic (23–25). These strategies require selectable markers, usually a drug-resistance gene or nutritional gene, to isolate cells that contain the desired construct. Recently, *in vitro* assembled CRISPR/Cas9 transformation systems have been utilized for highly efficient genetic manipulation of fungi (26–31). Due to its high efficiency, the CRISPR/Cas9 system has been adapted as a "marker-free" system to perform genome editing without needing a selectable marker (28).

Here, we adapted a protoplast-mediated transformation protocol to integrate a plasmid into the genome of *H. werneckii* ectopically. This ectopic integration strategy can be used to express the green fluorescent protein (GFP) in the cytoplasm of *H. werneckii*. We also develop a marker-free CRISPR/Cas9 protocol for knocking out genes in *H. werneckii*. These new strategies will further our understanding of this remarkable yeast's biology and our understanding of eukaryotes able to grow in extreme environments.

## MATERIALS AND METHODS

### Strains, media, and culture conditions

*H. werneckii* EXF-2000 was used for these studies. Yeast cells were grown on Glucose Minimal Media Agar (GMM [Dextrose 10 g/L, Trace Elements, Salt Solution]). Cells were grown at 30°C for 5 days unless otherwise specified. A more detailed description of growth media and buffers used can be found here: https://benchling.com/s/prt-zH21wPSfPKialp38RIDl?m=slm-eS0kPb33EH8i9TbXWQi1.

### Protoplast generation

*H. werneckii* cells were inoculated on GMM agar plates and incubated at 30°C for 5–7 days to create a lawn. Cells were harvested using a cell scraper and resuspended in 40 mL of osmotic media (1.2 M magnesium sulfate and 10 mM sodium phosphate buffer, pH 5.8) in a 50 mL conical tube, then centrifuged at $3,000 \times g$ for 10 min. The supernatant was discarded, and cells were resuspended in 40 mL osmotic media with 200 mg of Vinotaste and split into two 50 mL conical tubes (20 mL each). Cells were digested for 4–8 h at 30°C and shaking at 75 rpm. After digestion, the osmotic media containing Vinotaste and digested *H. werneckii* were gently overlayed with 10 mL of trapping buffer (0.6 M sorbitol and 0.1 M Tris-HCl pH 7). Tubes then were centrifuged at $2,500 \times g$ for 15 min at 4°C. The cloudy layer (protoplasts) at the interface of the osmotic media and trapping buffer was carefully moved into a new 15 mL conical centrifuge tube using a sterile plastic 1 mL transfer pipet. Protoplasts were washed by adding up to 15 mL of ice-cold Sorbitol-Tris-Calcium Chloride (STC) buffer (1.2 M sorbitol, 10 mM $CaCl_2$, and

10 mM Tris-HCl pH 7.5) to the 15 mL conical tube, then centrifuged at 2,500 × $g$ for 8 min. The supernatant was removed, and protoplasts were carefully resuspended with 1 mL of ice-cold STC buffer and counted using a hemocytometer.

## Polyethylene glycol-CaCl$_2$-mediated transformation of pUCGH plasmid

In total, 200 µL of protoplasts was transferred into a 1.5 mL microcentrifuge tube and incubated with 1–5 µg of the pUCGH plasmid, which contains the hygromycin resistance gene (*hph*) and the sequence encoding eGFP ([https://benchling.com/s/seq-HGwa-SyZj8IaGHGQfFKdU?m=slm-f1Vae2jxQfiCz0i0CKiq](https://benchling.com/s/seq-HGwa-SyZj8IaGHGQfFKdU?m=slm-f1Vae2jxQfiCz0i0CKiq)), for 50 min to 1 h on ice (32). About 1.25 mL of polyethylene glycol (PEG)-CaCl$_2$ (60% PEG3350, 10 mM CaCl$_2$, and 50 mM Tris-HCl pH 7.5) was added to the protoplasts and incubated for 20 min at room temperature. STC buffer was added to protoplasts to reach 3 mL, and 300 µL of the transformed protoplasts were plated on 10 Petri dishes containing 20 mL of SMM agar. Protoplasts were allowed to recover for 24 h at room temperature. After 24 h, the protoplasts-containing plates were overlaid with 10 mL of SMM top agar with 450 µg/mL of hygromycin B. Colonies appeared between 7 and 15 days of incubation. Individual colonies were transferred onto fresh GMM agar with hygromycin B (150 µg/mL) and incubated at 30°C.

## Imaging EXF-2000

A wet mount of each strain was made by smearing *H. werneckii* cells in 20 µL of distilled water. Cells were then imaged using a widefield microscope (Leica DMi8) using a 100× oil apochromat objective, and images were captured using a Leica K5 microscope camera.

## Assembly of CRISPR/Cas9 RNPs

In total, 5 µL of one crRNA (Table 1), tracrRNA, and nuclease-free duplex buffer (Integrated DNA Technology, IDT) were mixed and heated at 95°C for 5 min. Then, it was cool-down at room temperature to assemble the gRNA. 6 µL of each 33 µM gRNA, 6 µL of Cas9 nuclease (1 µg/µL) (Integrated DNA Technology, IDT), and 8.5 µL of Cas9 working buffer (20 mM HEPES and 150 mM KCl, pH 7.5) were mixed and incubated for 5 min at room temperature. CRISPR/Cas9 complex was osmotically stabilized by adding an equal volume of 2× STC buffer.

## Marker-free CRISPR/Cas9-mediated transformation

The osmotically stabilized CRISPR/Cas9 complex, 70 µL of protoplast, and 200 µL of PEG-CaCl$_2$ were gently mixed in a 50 mL conical tube. STC buffer instead of the osmotically stabilized CRISPR/Cas9 complex was used as a negative control. The protoplast mixture was incubated on ice for at least 30 min. 1 mL of PEG-CaCl$_2$ was added and incubated for 15 min at room temperature. 325 µL of the protoplasts were inoculated onto three large SMM agar plates (150 mm × 15 mm) and incubated overnight at room temperature. Then, plates were incubated at 30°C until colonies started appearing on the plates.

## PCR and sequencing of *alb1a* and *alb1b* locus

In total, 200 mg of yeast cells was harvested from GMM plates and placed in a 2 mL screw top tube containing 200 µL of 0.1 mm diameter soda lime glass beads. Then, cells were bead beaten for three cycles of 60 s, bead beating and 30 s resting. Bead beaten cells were resuspended in 500 µL CTAB, 20 µL RNAase A, and 40 µL of Proteinase K.

**TABLE 1** crRNAs used in this study

| crRNA | Target | Sequence |
| --- | --- | --- |
| alb1-gRNA-1 | 5′ of *alb1* | AAGCGCUUGACGGUGGUUGGCGG |
| alb1-gRNA-2 | 5´of *alb1* | AGCGUUGCGCGAAAGCGCUGCGG |

Resuspended cells were incubated for 1 h at 65°C. After incubation, cells were spun down for 10 min at 16,000 × *g*. DNA was purified using Promega's MAXWELL RSC PureFood GMO and Authentication Kit following the manufacturer's protocol. ProNex Size Selective Purification (Promega) was used to remove the occasional melanin carryover. Paralog-specific primers were designed to amplify and sequence the *alb1a* or *alb1b* genes (Table 2). Amplicons were run in 0.8% agarose gels for 45 min at 150 V. Bands were cut and purified using E.Z.N.A. Gel Extraction Kit (Omega Bio-Tek) following the manufacturer's protocol. Purified amplicons were sent to the Virginia Tech Genomics Sequencing Center for Sanger sequencing. Sequences were aligned to the reference EXF-2000 *alb1a* and *alb1b* sequence using local MAFFTv7 to identify mutations (33).

## RESULTS

### PEG calcium chloride protoplast-mediated transformation allows for ectopic integration of the pUCGH plasmid

*H. werneckii's* susceptibility to drugs is determined by the environment (3, 4). We noticed that *H. werneckii* is susceptible to hygromycin B when grown on glucose minimum media, the same media commonly used to grow *Aspergillus fumigatus* (29). Based on this, we used hygromycin B as our selectable marker and used a sorbitol-stabilized glucose minimum media (SMM) agar to perform our transformations. We consistently obtained protoplasts after 4–8 h of digesting the cell wall using Vinotaste (Novozymes). We then transformed the protoplast using the pUCGH vector, which contains the hygromycin B resistance gene (*hph*) under the control of *Aspergillus nidulans gpdA* promoter and eGFP under the control of the *Aspergillus oryzae tef1* promoter (32). We used a high quantity of plasmid to induce ectopic integration (1–5 µg) of the plasmid. Protoplasts were allowed to recover for 24 h before overlaying them with 10 mL of SMM top agar containing hygromycin B (450 µg/mL). Hygromycin B-resistant colonies emerged after 7 days of incubation at 30°C (Fig. 1A). Colonies were picked and streaked into a small petri dish containing GMM agar supplemented with hygromycin B (150 µg/mL). Transform-ants expressed eGFP even after five passages in GMM agar containing hygromycin B (150 µg/mL) (Fig. 1B). Thus, this approach can be utilized for ectopic integration of markers for cell biology studies.

### CRISPR/Cas9 marker-free transformation

More targeted genetic approaches are needed to further our understanding of *H. werneckii* biology. Similar to other ascomycetes, adapting the PEG-CaCl$_2$-mediated transformation protocol has proven challenging for targeted gene manipulation due to the low homologous recombination rate (34). This challenge is further exacerbated due to the EXF-2000 strain having a significant portion of its genome duplicated due to an intraspecific hybridization event (10). Due to these challenges, we decided to adopt a CRISPR/Cas9 marker-free approach and target the *H. werneckii* melanin synthesis pathway to have a visual phenotype we could screen (28). We identified two copies of the alb1 gene–*alb1a* (BTJ68_00107) and *alb1b* (BTJ68_01291)–using FungiDB (35). We targeted a conserved region between *H. werneckii's* two *alb1* paralogs using two gRNAs (Table 1). We generated protoplast using the same approach for the ectopic integration, and without standardizing the number of protoplasts, we obtained approximately three to five colonies that exhibited an albino phenotype (Fig. 2A) per transformation. We decided to determine if the concentration of protoplasts might impact the efficiency of

**TABLE 2** Primers used in this study

| Primer | Target | Sequence |
|---|---|---|
| alb1a-seq-F | *alb1a* | GACGGACGATCACAGCAATA |
| alb1a-seq-R | *alb1a* | GTTCCAATACGGTGGAGCTT |
| alb1b-seq-F | *alb1b* | TCACGAACTGAATACGGACG |
| alb1b-seq-R | *alb1b* | TACGATGGAGCTTGGTAGAC |

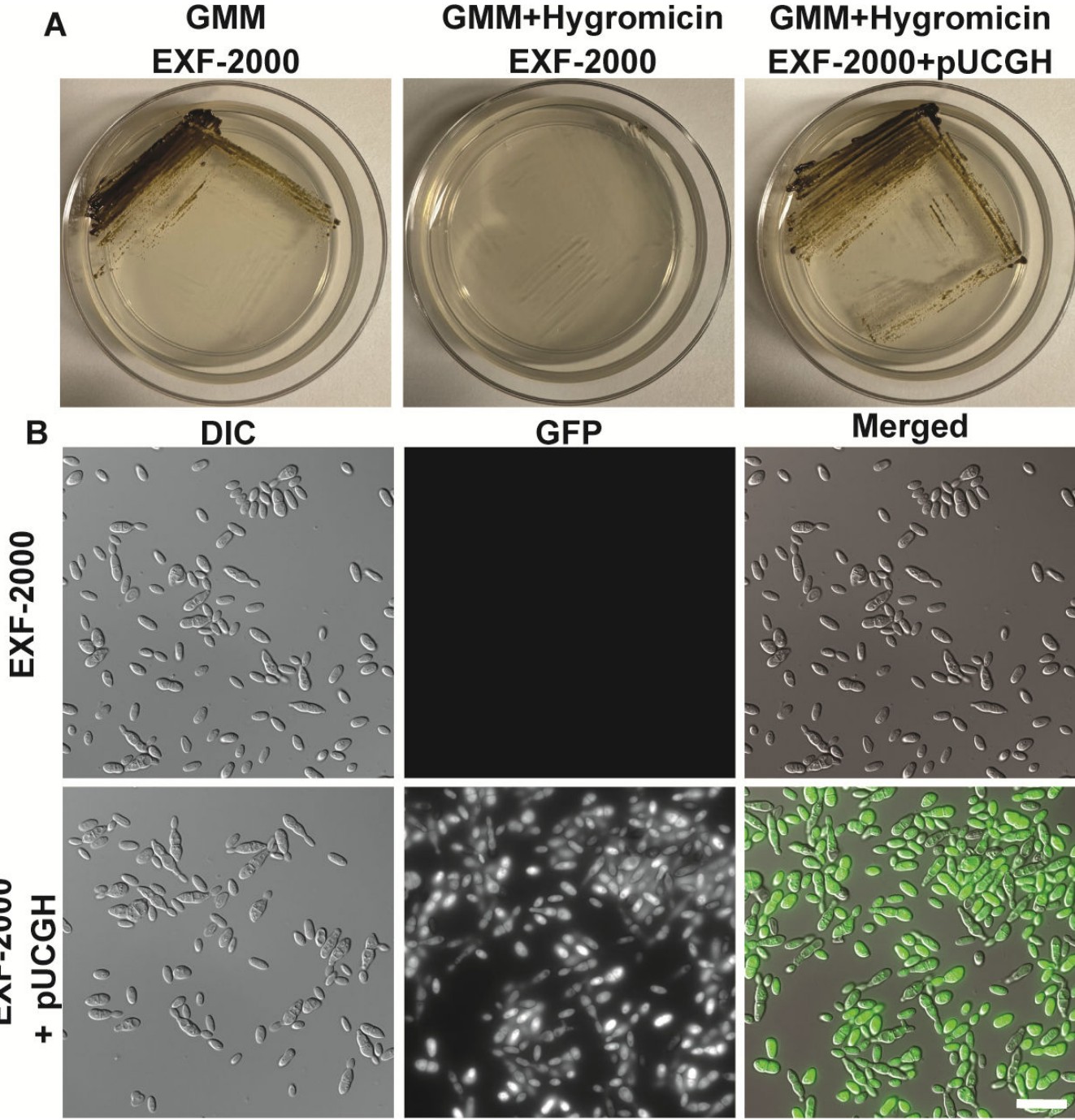

**FIG 1** Integration of the pUCGH plasmid in *H. werneckii*. (A) *H. werneckii* EXF-2000 is susceptible to hygromycin B in GMM media, and hygromycin B allows for the selection of ectopic integration of the pUCGH plasmid. Plates were streaked into the respective media, incubated at 37°C, and photographed after 5 days. (B) *H. werneckii* with integrated pUCGH expresses eGFP under the *A. oryzae tef1* promoter. Micrographs were obtained using a 100× objective. Scale bar, 20 µm.

our CRISPR/Cas9 approach (Fig. 2B). We observed that transforming a mix containing $10^4$ protoplasts led to approximately 6.5% success rate, compared to the ~1.9% success rate when transforming $10^5$ protoplasts. We did not observe any spontaneous albino mutant arising from the protoplast generation or the exposure of the protoplast to the STC buffer. The albino mutants could still not produce melanin after five passages, indicating that the deletion of the *alb1* genes was stable (Fig. 2C). These albino mutants retain the same cell morphology as the wild-type when grown in GMM agar (Fig.

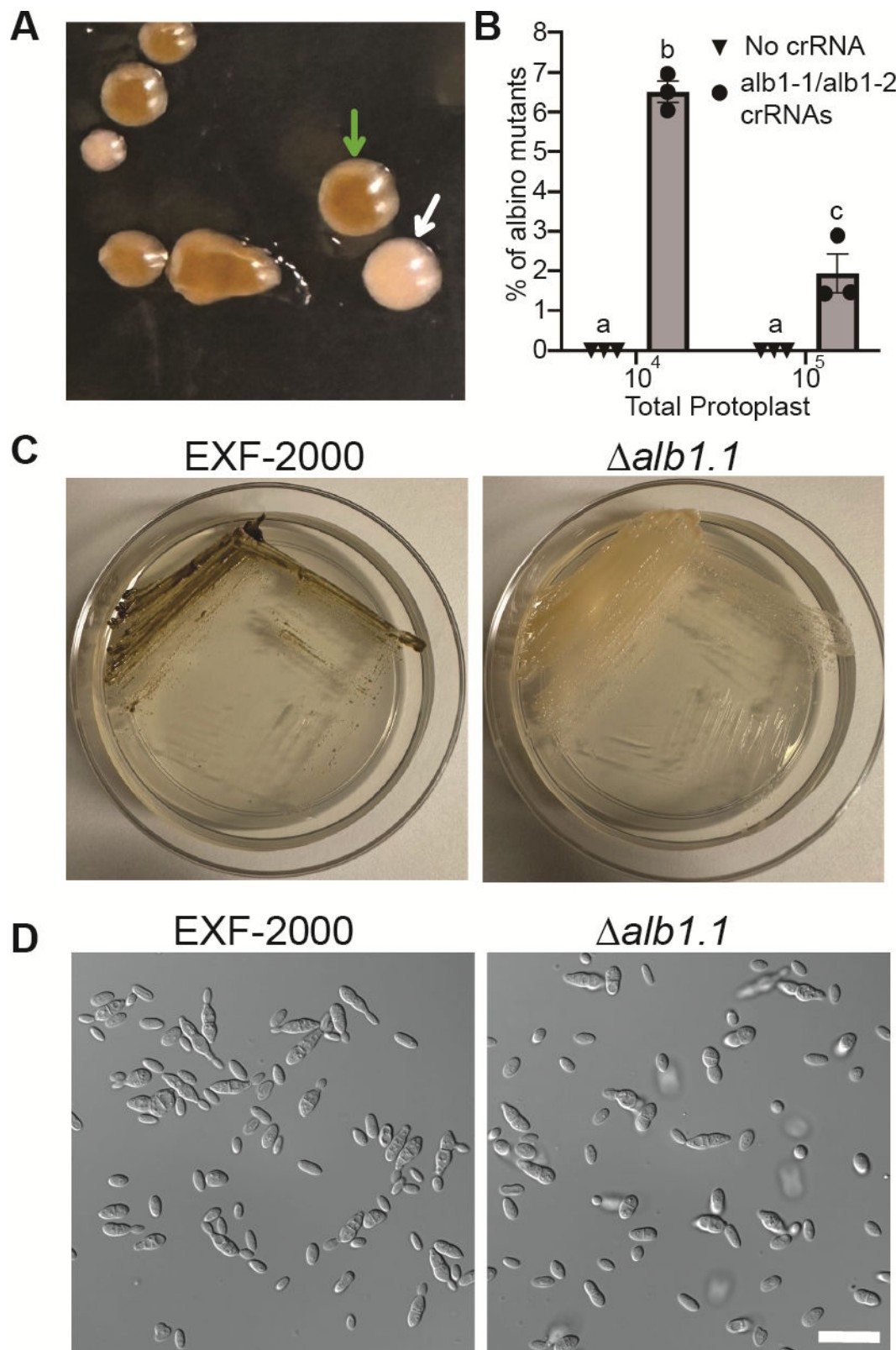

**FIG 2** Generation of albino mutants using CRISPR/Cas9. (A) Albino mutant in transformation plate (white arrow) next to melanized EXF-2000 (green arrow). (B) A lower amount of protoplast increases the efficiency of CRISPR/Cas9 transformation. $10^4$ or $10^5$ protoplasts were transformed with the CRISPR/Cas9 complex with each crRNA (circles) or STC buffer (inverted triangle). STC buffer-only protoplasts did not result in spontaneous albino mutants. Using $10^4$ protoplasts led to an

Fig 2 (Continued)

average of 6.5% albino mutants, in contrast to the 1.9% when using $10^5$ protoplasts. Two-way ANOVA followed by Tukey's *post hoc* test was used to compare the means of each experimental condition. Statistical significance was determined by $P \leq 0.05$. (C) The Δ*alb1.1* fails to melanize after prolonged incubation in GMM. Plates were streaked and incubated at 30°C for 5 days before being photographed. (D) Δ*alb1.1* has a similar morphology as the parent EXF-2000 strain when grown in GMM agar. Micrographs were obtained using a 100× objective. Scale bar, 20 µm.

2D), indicating that melanin does not impact cell morphology in these conditions. A 1.5 kb genomic region that covers the site targeted by CRISPR/Cas9 was amplified using primers specific to each *alb1* paralog (Table 2). Gel electrophoresis for the *alb1a* PCR reaction showed bands of similar size between the three independent mutants and the wild-type EXF-2000 strain, ~1.5 kb (Fig. 3A). Similarly, the Δ*alb1.1* and Δ*alb1.3* strains had *alb1b* amplicons of comparable size to the EXF-2000 band (Fig. 3B). In contrast, the Δ*alb1.2* strain had a smaller amplicon of ~1 kb, and this mutant had a deletion in their *alb1b* gene of ~500 bp. In the *alb1a* locus, we found that all the Δ*alb1* strains had similar mutations constrained between the Cas9 cut sites (Fig. 3C). The Δ*alb1.1* and Δ*alb1.3* followed a similar pattern for the *alb1b* locus (Fig. 3D). In concordance with the amplicon size, the Δ*alb1.2 alb1b* locus had a 528 bp deletion. The EXF-2000 *alb1a* and *alb1b* loci sequences are the same as the ones reported in the reference genome (11).

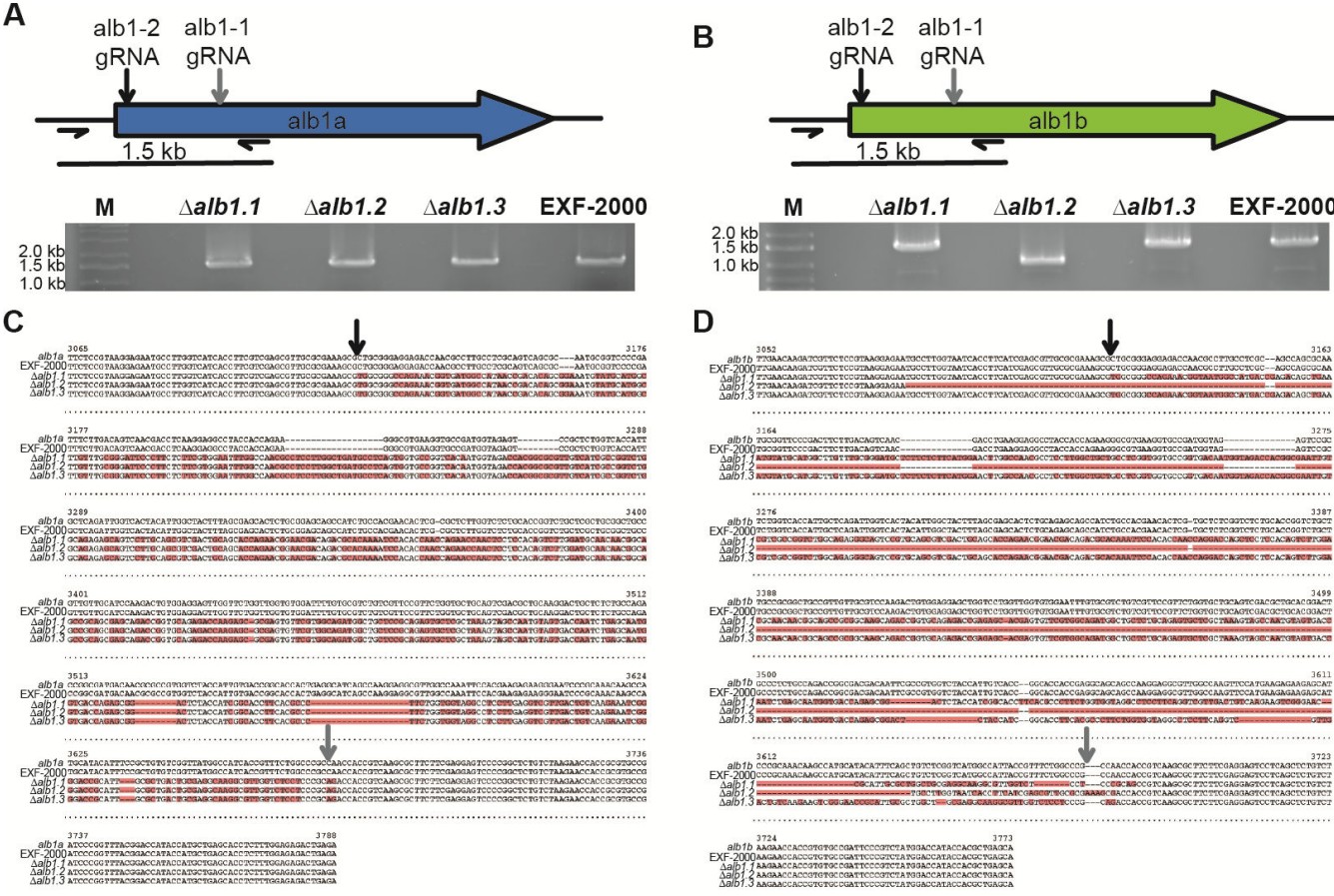

FIG 3 *alb1a* and *alb1b* loci are mutated in the Δ*alb1* strains. (A and B) Gel electrophoresis of the PCR amplicons of the *alb1a* (A) and *alb1b* (B) 5′ gene region. M, marker (GeneRuler 1 kb plus). (C and D) Alignment of the *alb1a* (C) and *alb1b* (D) loci. The gray arrow points at the expected cut site for Cas9 loaded with alb1-1 gRNA, and the black arrow points at the predicted cut site for Cas9 loaded with alb1-2 gRNA. Red highlights point to areas of divergence between the strains *alb1a* or *alb1b* sequence and the reference genome.

## DISCUSSION

*H. werneckii* has a unique cell cycle division and can grow at high salt concentrations (3, 4, 36). Due to most isolates being intraspecific hybrids with significant phenotypic variation, *H. werneckii* can become a model for understanding how hybridization leads to novel phenotypes (3, 9). However, the lack of genetic tools has limited the ability to study this organism. Here, we develop two different genetic transformation approaches to further our studies of this unconventional yeast. First, we have developed a plasmid-based ectopic integration protocol. This system can be used to express cellular markers for live-cell imaging. For example, we could use it to insert a histone-GFP plasmid and monitor nuclear dynamics. Second, we develop a marker-free CRISPR/Cas9 protocol to target specific genes.

One caveat is that our current CRISPR/Cas9 protocol still needs to be more effective and would not be practical for genes that would not have a visual readout. On average, we had a 6–7% success rate on targeting both gene paralogs, while other organisms had a 50–90% success rate (28–30). However, these organisms were haploid, and only one copy of the gene needed to be mutated, not the two copies of most genes in the intraspecific hybrids of *H. werneckii*. Finally, as there is a marker system and CRISPR/Cas9 can overcome the homologous recombination rate, it is possible to combine both methods to develop a marker-based CRISPR/Cas9 approach and select those mutants with successful integration of our cassette of interest (29). At the moment of this publication, only one marker has been developed (hygromycin) for *H. werneckii*. Thus, this approach might only be limited to protein tagging or deletion of genes with only one copy. Nonetheless, other drugs can be explored to expand the available range of selectable markers.

## ACKNOWLEDGMENTS

The authors thank Dr. William Steinbach and Dr. Praveen Juvaadi for sharing the pUCGH plasmid. The authors also want to thank Dr. Rachel Whitaker, Dr. Scott Dawson, and the Marine Biological Laboratory's (MBL) Microbial Diversity course for allowing us to use their microscope and resources. The author wants to thank Norman van Rhijn (Manchester Fungal Infection Group) and Robb Cramer's Lab (Dartmouth) for sharing their CRISPR/Cas9 protocols that were the basis for developing our protocol. This project was supported through the L. & A. Colwin Summer Research Fellowship awarded to J.V.M. J.V.M wants to thank William Wolf (Virginia Tech) and the Vargas-Muñiz Lab for critically reading this manuscript. Finally, the authors want to recognize FungiDB and VEuPathDB, which are invaluable resources for the community, and this project would not have been possible without their support and tools.

## AUTHOR AFFILIATIONS

[1]Department of Biological Sciences, Florida Gulf Coast University, Ft. Myers, Florida, USA

[2]Molecular Biology, Microbiology, and Biochemistry Program, School of Biological Science, Southern Illinois University, Carbondale, Illinois, USA

[3]Microbiology Program, School of Biological Sciences, Southern Illinois University, Carbondale, Illinois, USA

[4]Department of Biological Sciences, Virginia Tech, Blacksburg, Virginia, USA

[5]Fralin Life Science Institute, Virginia Tech, Blacksburg, Virginia, USA

[6]Center for Emerging, Zoonotic, and Arthropod-borne Pathogens, Virginia Tech, Blacksburg, Virginia, USA

[7]Early Career Whitman Fellow, Marine Biological Laboratory, Woods Hole, Massachusetts, USA

## AUTHOR ORCIDs

José M. Vargas-Muñiz  http://orcid.org/0000-0003-0138-8660

## AUTHOR CONTRIBUTIONS

Yainitza Hernandez-Rodriguez, Conceptualization, Investigation, Methodology, Writing – review and editing | A. Makenzie Bullard, Investigation, Methodology, Writing – review and editing | Rebecca J. Busch, Investigation, Methodology, Writing – review and editing | Aidan Marshall, Investigation, Methodology, Validation | José M. Vargas-Muñiz, Conceptualization, Formal analysis, Funding acquisition, Investigation, Methodology, Project administration, Resources, Supervision, Writing – original draft, Writing – review and editing

## ADDITIONAL FILES

The following material is available online.

### Open Peer Review

**PEER REVIEW HISTORY (review-history.pdf).** An accounting of the reviewer comments and feedback.

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
