## [Reviewer comments · Microbiology Spectrum]

Microbiology Spectrum

Strategies for Genetic Manipulation of the Halotolerant Black Yeast *Hortaea werneckii*: Ectopic DNA Integration and Marker-free CRISPR/Cas9 Transformation

Yainitza Hernandez-Rodriguez, A. Makenzie Bullard, Rebecca Busch, Aidan Marshall, and Jose Vargas-Muniz

Corresponding Author(s): Jose Vargas-Muniz, Virginia Polytechnic Institute and State University

Review Timeline:

Submission Date:	September 27, 2024
Editorial Decision:	October 29, 2024
Revision Received:	November 12, 2024
Accepted:	November 18, 2024

Editor: Miguel Penalva

Reviewer(s): The reviewers have opted to remain anonymous.

Transaction Report:

DOI: <https://doi.org/10.1128/spectrum.02430-24>

Re: Spectrum02430-24 (Strategies for Genetic Manipulation of the Halotolerant Black Yeast *Hortaea werneckii*: Ectopic DNA Integration and Marker-free CRISPR/Cas9 Transformation)

Dear Dr. Jose Vargas-Muniz:

I have now collected two reviews for your manuscript. As you may see, both reviews are positive, but the referees would be willing to get a more detailed account of the Methodology. Please introduce the requested information and the minor edits and return the membership to me at your earliest convenience.

In any case please return the manuscript within 60 days; if you cannot complete the modification within this time period, please contact me. If you do not wish to modify the manuscript and prefer to submit it to another journal, notify me immediately so that the manuscript may be formally withdrawn from consideration by Spectrum.

Revision Guidelines

Sincerely,
Miguel Penalva
Editor
Microbiology Spectrum

Reviewer #1 (Comments for the Author):

The overall manuscript is well written and the objective of transforming the black yeast *Hortaea werneckii* is achieved with success. My only question is about the composition of PEG-CaCl₂ solution, which is not described. Type and percentage of PEG are crucial and this has to be fully described for reproducibility.

Reviewer #2 (Comments for the Author):

Hernandez-Rodriguez et al. report on the methodology adapted for genetic manipulation of a black yeast *Hortaea werneckii*, including a marker-free transformation of this yeast. It is, to the best of my knowledge, the first such successful attempt with this salt-tolerant species, and is as such highly relevant for continuing research in the field.

The design of the study is sound, the manuscript is concise, and the results clearly presented. However, some improvements should be made before the manuscript is accepted for publication.

First and foremost, the methods section should be developed further. The language should be carefully edited and, more importantly, enough information should be provided to support reproducibility. Since this is a methodological article, the methodology section should be written most rigorously of all - but is instead the weakest part of the text in its current form.

In other parts of the text, there are some remaining small typographical errors throughout the text. I list some below, but I am not qualified to do comprehensive editing.

While the topic is very specialized, I believe it is an important contribution to the growing genetic toolbox in the field of non-conventional yeasts and I encourage the authors to improve this report on an otherwise very good study in order to make it as useful as possible to others in the field.

Some specific and minor comments are listed below.

Specific comments:

53: genes instead of gene

64: *H. werneckii* grows at around 30% NaCl (w/v), but generally not in saturated solutions; there are some (mostly older) reports about saturated limit, but in newer studies it is mostly recognised that while the yeast comes close to saturation limit, it does not actually grow in saturated solutions

74-75: Not all isolates are hybrids - many isolates are simply haploid, please check the reference 12 for more detail

84: Agrobacterium-mediated

84-85: transformation of chemically competent cells is not listed, although it is arguably the most commonly used method, given its popularity in *S. cerevisiae* and some other species

97: missing fullstop

104: on GMM agar plates

106: use past tense as elsewhere

107: density of culture? How many cells per tube?

111: conical what? Tube?

111 what does STC stand for?

115 protoplasts were (plural) - correct that in the following sentences as well

115 1.5 mL what?

116 what is a pUCGH plasmid? What does it contain?

119 with what volume of SMM top agar?

121 onto agar, not into - correct that in other places as well

136 use a space between the value and units in all cases (70 μ L, not 70 μ L) - the text is missing spaces in many places

152 what was used to purify DNA from the gel?

154 sequences were aligned to what? The reference? Each other?

164 Vinotaste - use capitalised

166 resistance, not resistant (the strain may be resistant, but the gene is not)

183 missing space after "screen"

184 missing space after FungiDB

226 Cas9, not Ca9

November 11, 2024

Dr. Miguel Penalva
Editor
Microbiology Spectrum

RE: Revise Manuscript Spectrum02430-24; “Strategies for Genetic Manipulation of the Halotolerant Black Yeast *Hortaea werneckii*: Ectopic DNA Integration and Marker-free CRISPR/Cas9 Transformation.”

Dear Dr. Miguel Penalva,

We appreciate the opportunity to revise our papers in response to the reviewer’s critiques. We have addressed their comments and incorporated language suggestions and clarifications requested by the reviewers. Changes to the manuscript can be found in the marked-up document, and a “clean” version has also been submitted. Below, you will find our point-by-point response (in blue) to the questions and suggestions provided by the reviewers.

REVIEWER 1:

The overall manuscript is well written and the objective of transforming the black yeast *Hortaea werneckii* is achieved with success. My only question is about the composition of PEG-CaCl₂ solution, which is not described. Type and percentage of PEG are crucial and this has to be fully described for reproducibility.

Response: We appreciate the overall positive assessment of our manuscript. As requested, we added the composition of the PEG-CaCl₂ solution (60% PEG3350, 10 mM CaCl₂, 50 mM Tris-HCl pH 7.5) (LN 139-140). We also included a link to all the media and buffer recipes used in this manuscript(<https://benchling.com/s/prt-zH21wPSfPKialp38RIDI?m=slm-eS0kPb33EH8i9TbXWQi1>) (LN109-111)

REVIEWER 2:

Hernandez-Rodriguez et al. report on the methodology adapted for genetic manipulation of a black yeast *Hortaea werneckii*, including a marker-free transformation of this yeast. It is, to the best of my knowledge, the first such successful attempt with this salt-tolerant species, and is as such highly relevant for continuing research in the field. The design of the study is sound, the manuscript is concise, and the results clearly presented. However, some improvements should be made before the manuscript is accepted for publication. First and foremost, the methods section should be developed further. The language should be carefully edited and, more importantly,

enough information should be provided to support reproducibility. Since this is a methodological article, the methodology section should be written most rigorously of all - but is instead the weakest part of the text in its current form.

In other parts of the text, there are some remaining small typographical errors throughout the text. I list some below, but I am not qualified to do comprehensive editing. While the topic is very specialized, I believe it is an important contribution to the growing genetic toolbox in the field of non-conventional yeasts and I encourage the authors to improve this report on an otherwise very good study in order to make it as useful as possible to others in the field. Some specific and minor comments are listed below.

Response: We appreciate the positive evaluation of our work. We agree with the reviewer's comment that the methodology section should be more detailed to allow reproducibility. We have added more details to our methods section in response to this weakness in our original submission. We also apologize for the typographical errors that might be a distraction from our manuscript. We have edited the manuscript to address those errors.

Specific comments:

53: genes instead of gene

Response: We incorporated the suggested edits to the manuscript.

64: *H. werneckii* grows at around 30% NaCl (w/v), but generally not in saturated solutions; there are some (mostly older) reports about saturated limit, but in newer studies it is mostly recognised that while the yeast comes close to saturation limit, it does not actually grow in saturated solutions

Response: Thank you for bringing this to our attention. We decided to remove that statement as it is redundant with the first sentence and does not quite fit the description of the second sentence where it is found.

74-75: Not all isolates are hybrids - many isolates are simply haploid, please check the reference 12 for more detail

Response: We apologize for this oversight. We rewrote that sentence to incorporate the haploids (LN 71-78).

84: Agrobacterium-mediated

Response: We incorporated the suggested edits to the manuscript.

84-85: transformation of chemically competent cells is not listed, although it is arguably the most commonly used method, given its popularity in *S. cerevisiae* and some other species

Response: We included lithium acetate chemically competent transformation (LN88-89, Ref 26)

97: missing fullstop

Response: We incorporated the suggested edits to the manuscript.

104: on GMM agar plates

Response: We incorporated the suggested edits to the manuscript.

106: use past tense as elsewhere

Response: We incorporated the suggested edits to the manuscript.

107: density of culture? How many cells per tube?

Response: We did not quantify how many cells were per tube, nor did we measure the density of the culture. Nonetheless, we added details on how cells were harvested and digested (LN113-118). Briefly, we generated a lawn of EXF-2000 on GMM-Agar by growing the cells at 30°C for 5-7 days. Cells were harvested using a cell scraper and washed 40 ml osmotic media, followed by digestion with Vinotaste for 4-8 hours. Using this approach consistently yielded an abundant yield of EXF-2000 protoplast.

111: conical what? Tube?

Response: We apologize for the lack of clarity. We meant conical tubes.

111 what does STC stand for?

Response: STC stands for Sorbitol Tris CaCl₂. We have added this information to the manuscript (LN124-125)

115 protoplasts were (plural) - correct that in the following sentences as well

Response: We incorporated the suggested edits to the manuscript.

115 1.5 mL what?

Response: We apologize for that omission; it should read 1.5 ml microcentrifuge tube.

116 what is a pUCGH plasmid? What does it contain?

Response: The pUCGH plasmid is a previously published plasmid commonly used in *Aspergillus fumigatus*, first reported in Ref (34). This plasmid contains the *hph* hygromycin B resistance gene under the control of *Aspergillus nidulans gpdA* promoter and eGFP under the *Aspergillus*

oryzae tef1 promoter. This information was added on LN140-142 and LN212-214. We also included a link to the plasmid map (<https://benchling.com/s/seq-HGwaSyZj8IaGHGQfFKdU?m=sIm-flVae2jxQfiCz0i0CKiq>)

119 with what volume of SMM top agar?

Response: We added the volume of SMM top agar (10 ml per plate). We also added the volume of SMM agar used in each petri dish, as this information is relevant for this protocol.

121 onto agar, not into - correct that in other places as well

Response: We incorporated the suggested edits to the manuscript.

136 use a space between the value and units in all cases (70 μ L, not 70 μ L) - the text is missing spaces in many places

Response: We incorporated the suggested edits to the manuscript.

152 what was used to purify DNA from the gel?

Response: We use the Omega Bio-Tek E.Z.N.A. Gel Extraction Kit to purify DNA from the DNA electrophoresis gel.

154 sequences were aligned to what? The reference? Each other?

Response: We apologize for the lack of detail in this sentence. We aligned the sequences to the reference EXF-2000 *alb1a* and *alb1b* sequences. The new sentence reads as follows: "Sequences were aligned to the reference EXF-2000 *alb1a* and *alb1b* sequence using local MAFFT v7 to identify mutations".

164 Vinotaste - use capitalized

Response: We incorporated the suggested edits to the manuscript.

166 resistance, not resistant (the strain may be resistant, but the gene is not)

Response: We incorporated the suggested edits to the manuscript.

183 missing space after "screen"

Response: We incorporated the suggested edits to the manuscript.

184 missing space after FungiDB

Response: We incorporated the suggested edits to the manuscript.

226 Cas9, not Ca9

Response: We incorporated the suggested edits to the manuscript.

In summary, we believe that we have responded to the reviewer's suggestions with this revision. We are excited about the positive reception of our work, and by addressing the points the reviewers brought up, our revised manuscript is stronger and achieves the goals of a methods paper.

Sincerely,

José Vargas-Muñiz, Ph.D.
Assistant Professor
Department of Biological Sciences
Virginia Tech
Blacksburg, VA 24061

Re: Spectrum02430-24R1 (Strategies for Genetic Manipulation of the Halotolerant Black Yeast *Hortaea werneckii*: Ectopic DNA Integration and Marker-free CRISPR/Cas9 Transformation)

Dear Dr. Jose Vargas-Muniz:

I am pleased to inform you that your manuscript has been accepted, and I am forwarding it to the ASM production staff for publication. Your paper will first be checked to make sure all elements meet the technical requirements. ASM staff will contact you if anything needs to be revised before copyediting and production can begin. Otherwise, you will be notified when your proofs are ready to be viewed.

Sincerely,
Miguel Penalva
Editor
Microbiology Spectrum